# Generative Modeling for Protein Structures

**Namrata Anand**
Bioengineering Department, Stanford
namrataa@stanford.edu

**Po-Ssu Huang**
Bioengineering Department, Stanford
possu@stanford.edu

## Abstract

Analyzing the structure and function of proteins is a key part of understanding biology at the molecular and cellular level. In addition, a major engineering challenge is to design new proteins in a principled and methodical way. Current computational modeling methods for protein design are slow and often require human oversight and intervention. Here, we apply Generative Adversarial Networks (GANs) to the task of generating protein structures, toward application in fast *de novo* protein design. We encode protein structures in terms of pairwise distances between $\alpha$-carbons on the protein backbone, which eliminates the need for the generative model to learn translational and rotational symmetries. We then introduce a convex formulation of corruption-robust 3D structure recovery to fold the protein structures from generated pairwise distance maps, and solve these problems using the Alternating Direction Method of Multipliers. We test the effectiveness of our models by predicting completions of corrupted protein structures and show that the method is capable of quickly producing structurally plausible solutions.

## 1 Introduction

The ability to determine and design protein structures has deepened our understanding of biology. Advancements in computational modeling methods have led to remarkable outcomes in protein design including the development of new therapies [1, 2], enzymes [3, 4, 5], small-molecule binders [6], and biosensors [7]. These efforts are largely limited to modifying naturally occurring, or "native," proteins. To fully control the structure and function of engineered proteins, it is ideal in practice to create proteins *de novo* [8]. A fundamental question is discovering new, non-native folds or structural elements that can be used for designing these novel proteins. The protein design problem remains a major engineering challenge because the current design process relies heavily on heuristics, requiring subjective expertise to negotiate pitfalls that result from optimizing imperfect scoring functions.

We demonstrate the potential of deep generative modeling for fast generation of new, viable protein structures for use in protein design applications. We use Generative Adversarial Networks (GANs) to generate novel protein structures [9, 10] and use our trained models to predict missing sections of corrupted protein structures. We use a data representation restricted to structural information–pairwise distances of $\alpha$-carbons on the protein backbone. Despite this reduced representation, our method successfully learns to generate new structures and, importantly, can be used to infer solutions for completing corrupted structures. We use the Alternating Direction Method of Multipliers (ADMM) algorithm to "fold" 2D pairwise distances into 3D Cartesian coordinates [11]. The algorithm presented is a new method to do 3D structure generation and recovery using deep generative models, which is invariant to transformations in the Lie group of rotations and translations (SE(3)).

This paper is a step toward *learning* the protein design and folding process. Ultimately, our goal is to extend the generative model described, with subsequent steps of reinforcement learning or imitation learning to produce realistic protein structure at atomic resolution.

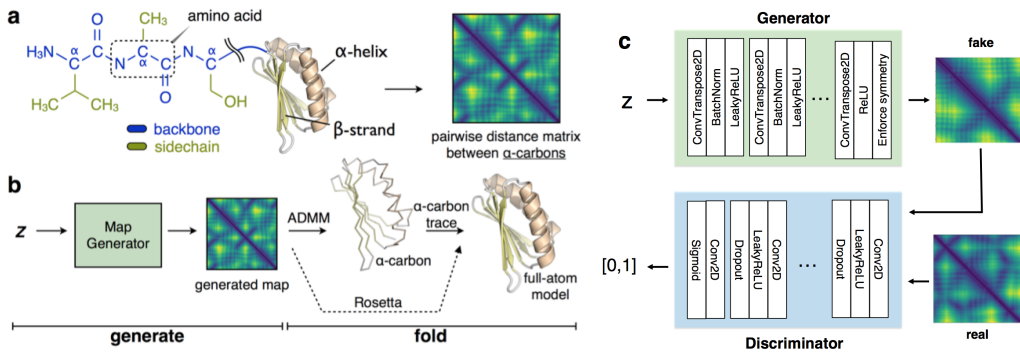

Figure 1: **a) Data representation**. Proteins are made up of chains of amino acids and have secondary structure features such as alpha helices and beta sheets. We represent protein structures using pairwise distances in angstroms between the $\alpha$-carbons on the protein backbone. **b) Pipeline**. GAN generates a pairwise distance matrix, which is "folded" into a 3D structure by ADMM to get $\alpha$-carbon coordinate positions; a fast 'trace' script then traces a reasonable protein backbone through the $\alpha$-carbon positions. We also fold structures directly from pairwise distances using Rosetta (fragment sampling subject to distance constraints) **c) Model**. DCGAN model architecture used for generating pairwise distance maps. The generator takes in random vector $\mathbf{z} \sim \mathcal{N}(0, I)$ and outputs a fake distance map to fool the discriminator. The discriminator predicts whether inputs are real (data samples) or fake (generator output).

The main contributions of this paper are (i) a generative model of proteins that estimates the distribution of their structures in a way that is invariant to rotational and translational symmetry and (ii) a convex formulation for the resulting reconstruction problem that we show scales to large problem instances.

# 2 Background

## 2.1 Protein structure and design

Proteins are macromolecules made up of chains of amino acids, with side-chain groups branching off a connected backbone (Figure 1a). Interactions between the side-chains, the protein backbone, and the environment give rise to local secondary structure elements – such as helices, strands, or random coils – and to the ultimate 3D structure of the protein. The large number of possible conformations of the peptide backbone, as well as the requirement to satisfy correct chemical bonding geometry and interactions, makes the protein structural modeling problem challenging.

In this paper, we study sequence-agnostic structure generation; this is different from the task of protein structure prediction, in which the structure of the protein is predicted given the amino acid sequence. Although protein structures are determined by their primary amino acid sequence, in recent years, it has become more apparent that protein structures and protein-protein interfaces conform largely to structural motifs [12]. A well known example is helical coiled-coils, where the angles between two interacting and sequence diverse helices fall within a range to facilitate knobs-into-holes packing [13]. These observations emphasize the importance of understanding sequence agnostic backbone behaviors. Here, our goal is to try to sample from the distribution of viable protein backbones.

The conventional protein design process starts with designing a peptide backbone structure, which can either be derived from a native protein or artificially created (i.e. *de novo* design); this is followed by finding the sequence of amino acids or side-chain residues which will fold into the backbone structure. Often, a structure is only partially modified such that one or more segments are manipulated while the rest of the structure is kept intact; this is referred to as a *loop modeling* or *loop closure* problem. Ensuring that the resulting structure has a fully connected and plausible backbone can be difficult.

The current state-of-the-art method for designing protein structures is the Rosetta modeling suite. Guided by a heuristic energy function, Rosetta samples native protein fragments to fold backbones and can optimize over amino acid types and orientations to determine the most likely amino acid sequence corresponding to the designed backbone [14]. For loop modeling problems, Rosetta supplements fragment sampling with closure algorithms such as Cyclic Coordinate Descent (CCD) [15] and Kinematic Closure (KIC) [16] to ensure backbone connectivity. Hallmarks of the Rosetta methodology are that it has a highly refined energy function to guide sampling and its model building processes are intuitive and flexible. The main drawbacks of this method are that the fragment sampling step is very slow and the method requires extensive sampling steps before arriving at reasonable solutions. We suggest a fast method for loop modeling using a generative model, which takes into account the global structure of the protein.

## 2.2 Generative models

Generative Adversarial Networks (GANs) are a powerful class of models for generating new data samples that match the statistics of an input dataset [9]. GANs are made up of a generator network which seeks to produce realistic data and a discriminator network which tries to distinguish fake samples produced by the generator from real samples from the input dataset.

Given data $\mathbf{x}$ and random vector $\mathbf{z} \sim \mathcal{N}(0, I)$, the discriminator $D$ and generator $G$ each seek to maximize the following objectives

$$\max_D \ \mathbb{E}_{\mathbf{x} \sim p_{\text{data}}(\mathbf{x})}[\log D(\mathbf{x})] + \mathbb{E}_{\mathbf{z} \sim p_{\mathbf{z}}(\mathbf{z})}[\log(1 - (D(G(\mathbf{z}))))]$$
$$\max_G \ \mathbb{E}_{\mathbf{z} \sim p_{\mathbf{z}}(\mathbf{z})}[\log(D(G(z)))]$$

(1)

We use a deep convolutional generative adversarial network (DCGANs) as our generative model in this paper [10].

## 2.3 Related Work

Other than Rosetta-based fragment sampling methods, related work on sequence-agnostic generative models for protein backbones include TorusDBN [17] and FB5-HMM [18] which are Hidden Markov Models (HMMs) trained to generate local backbone torsion angles and $\alpha$-carbon coordinate placement, respectively. We baseline our work with these methods.

Indirectly related papers use neural network models to predict properties of and generate new small molecules and protein/DNA sequences. Some of these use neural networks on graph or string representations of small molecules [19, 20, 21, 22]. A recent paper uses deep neural networks to predict amino acid sequence for a corresponding structure [23]. Another result uses GANs to generate DNA sequences [24].

Structure prediction methods include residue coevolution-based structure prediction [25, 26] and recent work on neural network based methods [27]. These approaches assume the underlying amino-acid sequence is known.

We use ADMM to infer 3D coordinates from pairwise distances; similarly, others have used a semidefinite program (SDP) to infer protein structure from nuclear magnetic resonance (NMR) data, [28], using semidefinite facial reduction to reduce the size of the SDP.

The algorithm presented in this paper is a new method to do 3D structure generation and recovery using deep generative models in a manner invariant to transformations in SE(3). This method works because of the fixed order of the peptide chain. Current methods for 3D structure generation are not SE(3) invariant. A representative example is [29] who use GANs to produce structures in 3D voxel space; we baseline our method with this model.

## 3 Methods

### 3.1 Dataset and map generation

An overview of our method is shown in Figure 1b. We use data from the Protein Data Bank [30], a repository of experimentally determined structures available on-line. Although full-atom, high

resolution structures are available for use, we sought a representation of protein structure that would eliminate the need for explicitly encoding SE(3) invariance of 3D structures. We chose to encode 3D structure as 2D pairwise distances between $\alpha$-carbons on the protein backbone. This representation does not preserve information about the protein sequence (side chains) or the torsion angles of the polypeptide backbone, but preserves enough information to allow for structure recovery. We refer henceforth to these pairwise $\alpha$-carbon distance matrices as "maps." Note that the maps preserve the order of the peptide chain from N- to C- terminus and are SE(3) invariant representations of the 3D structure by construction.

To minimize structural homology between the GAN training data and the test data, we separated train and test structures by SCOP (Structural Classification of Proteins) fold type. We include an exact list of train and test set PDB IDs in the supplementary material. Note that the GAN does not require a train/test split, but our subsequent corruption recovery experiments do. To create our datasets, we extract non-overlapping fragments of lengths 16, 64, and 128 from chain 'A' for each protein structure starting at the first residue and calculate the pairwise distance matrices from the $\alpha$-carbon coordinate positions. Importantly, the inputs to the model are not all independently folded domains, but include fragments. We made this choice because we were unable to stably train a generative model for arbitrary length input structures. Our model for the scope of this paper is not necessarily learning protein structures which will fold, but rather learning building block features that define secondary and tertiary structural elements.

We generated 16-, 64-, and 128-residue maps by training GANs on the corresponding maps in our dataset. The model architecture is represented in Figure 1c. Experiment details are given in the supplementary material, Section A.1.

## 3.2 Folding generated maps

After generating pairwise distance maps, we must recover or "fold" the corresponding 3D structure. We tested two methods for folding generated maps. The first is using Rosetta's optimization toolkit to find a low-energy structure via fragment sampling, given distance constraints with slack. In practice, this takes several minutes to fold small structures of less than 100 residues because of the fragment and rotamer sampling steps (Figure S1a). This is is not a scalable method for folding and evaluating many structures; therefore we sought another, faster way to reconstruct 3D protein structure via the Alternating Direction Method of Multipliers (ADMM) algorithm [11].

### 3.2.1 ADMM

The task of determining 3D cartesian coordinates given pairwise distance measurements is already well understood and has a natural formulation as a convex problem [31]. Given $m$ coordinates $[a_1, a_2, \ldots a_m] = A \in \mathbb{R}^{n \times m}$, we form the Gram matrix $G = A^T A \in \mathcal{S}_+^m$. Note that $G$ is symmetric, positive-semidefinite with rank at most $n$. We want to recover $A$ given pairwise distance measurements $D$, with $d_{ij} = \|a_i - a_j\|_2$. Since $g_{ij} = a_i^T a_j$ and $d_{ij}^2 = g_{ii} + g_{jj} - 2g_{ij}$, we can find $G$ by solving an SDP over the positive semidefinite cone.

$$\min_{G, \eta} \lambda \|\eta\|_1 + \frac{1}{2} \sum_{i=1, j=1}^{m} (g_{ii} + g_{jj} - 2g_{ij} + \eta_{ij} - d_{ij}^2)^2 \tag{2}$$
$$\text{subject to} \quad G \in \mathcal{S}_+^n$$

where we have allowed a slack term $\eta$ on each distance measurement, whose $\ell_1$ norm is penalized, with weight $\lambda \in \mathbb{R}$. This slack term allows the model to be robust to sparse corruptions of distance measurements. This penalty is common in corruption-robust modeling approaches and has theoretical guarantees in other applications like Robust Principal Components Analysis [32]. We do not address such theoretical guarantees in this work but demonstrate its empirical properties in the supplement Section A.2.

While this optimization problem can be solved quickly using SDP solvers for systems where $n$ is small, the runtime of traditional solvers is quadratic in $n$ and renders large structure recovery problems out of reach. We found that indirect solvers like SCS were not able to handle problems with $n > 115$ (Figure S1b) [33, 34].

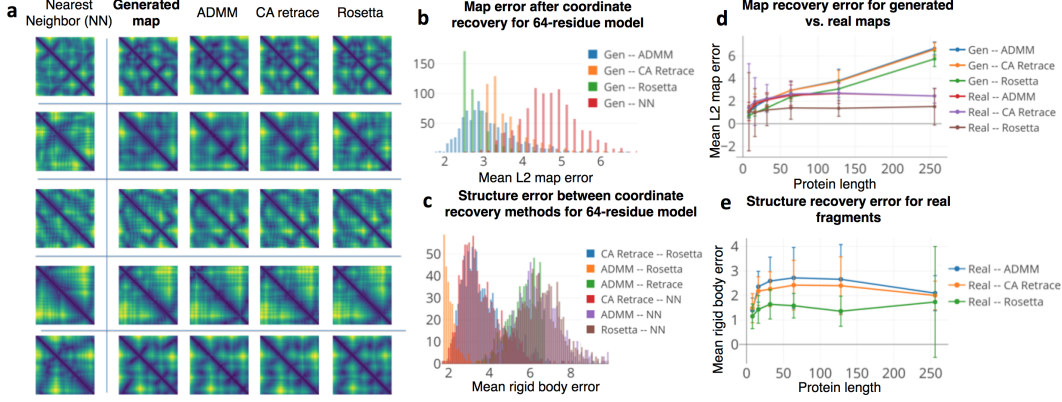

Figure 2: **a) Generated pairwise distance maps for the 64-residue model**, along with corresponding nearest neighbors (NNs) by $\ell_2$ distance in training dataset and maps after ADMM coordinate recovery, subsequent $\alpha$-carbon retrace step (CA retrace), and coordinate recovery by Rosetta fragment sampling. **b)** Distribution of $\ell_2$ map errors after coordinate recovery for generated maps ($n = 1000$). **c)** Distribution of $\alpha$-carbon rigid body alignment errors between folding methods for generated maps ($n = 1000$). **d)** Mean $\ell_2$ map errors after coordinate recovery for generated and real maps vs. protein length **e)** Mean $\alpha$-carbon rigid body alignment errors vs. protein length ($n = 64$ per data point).

Hence, we use ADMM which we found in practice converges to the correct solution quickly (Figure S1c). ADMM is a combination of dual ascent with decomposition and the method of multipliers. We write a modified optimization problem as

$$\min_{G,Z,\eta} \ \lambda \left\| \eta \right\|_1 + \frac{1}{2} \left( \sum_{i=1,j=1}^{m} (g_{ii} + g_{jj} - 2g_{ij} + \eta_{ij} - d_{ij}^2)^2 \right) + \mathbb{1}\{Z \in \mathcal{S}_+^m\} \tag{3}$$

$$\text{subject to } G - Z = 0$$

Now we decompose the problem into iterative updates over variables $G$, $Z$, and $U$ as

$$G_{k+1}, \eta_{k+1} = \underset{G,\eta}{\operatorname{argmin}} \ [\ \lambda \left\| \eta \right\|_1 + \quad \frac{1}{2} \sum_{i=1,j=1}^{m} (g_{ii} + g_{jj} - 2g_{ij} + \eta_{ij} - d_{ij}^2)^2 + \frac{\rho}{2} \left\| G - Z_k + U_k \right\|_2^2 \ ]$$

$$Z_{k+1} = \Pi_{\mathcal{S}_+^n}(G_{k+1} + U_k)$$

$$U_{k+1} = U_k + G_{k+1} - Z_{k+1}$$

$$\tag{4}$$

with augmented Lagrangian penalty $\rho > 0$. The update for $Z$ is simply the projection onto the set of symmetric positive semidefinite matrices of rank $n$. We find $G_k$ and $\eta_k$ by several iterations of gradient descent. After convergence, coordinates $A$ can be recovered from $G$ via SVD. Note that this algorithm is generally applicable to any problem for structure recovery from pairwise distance measurements, not only for protein structures. In practice, this algorithm is fairly robust to corruption of data (Section A.2).

Since our data representation only includes pairwise distances between $\alpha$-carbons, we need a fast method to recover the entire protein backbone given the $\alpha$-carbon coordinates outputted by ADMM. To do this, we use Rosetta to do local fragment sampling for every five carbons, constraining the original placement of the carbons. The backbone is joined by selecting the middle residue for each carbon 5-mer. This is followed by a short design procedure which finds a low-energy sequence for the designed backbone. This is the "$\alpha$-carbon trace" block shown in Figure 1b. Unlike running the full Rosetta protocol (dashed line, Figure 1b), this is a short procedure that runs in no more than a few minutes for large structures (Figure S1d).

The benefit of this procedure is that we recover realistic protein structures that adhere to the $\alpha$-carbon placement determined by ADMM, while only sampling 5-mers and not all native fragments; hence, the procedure runs an order of magnitude faster than the Rosetta protocol described above. The primary drawback is that the ADMM procedure cannot always correct for local errors in secondary

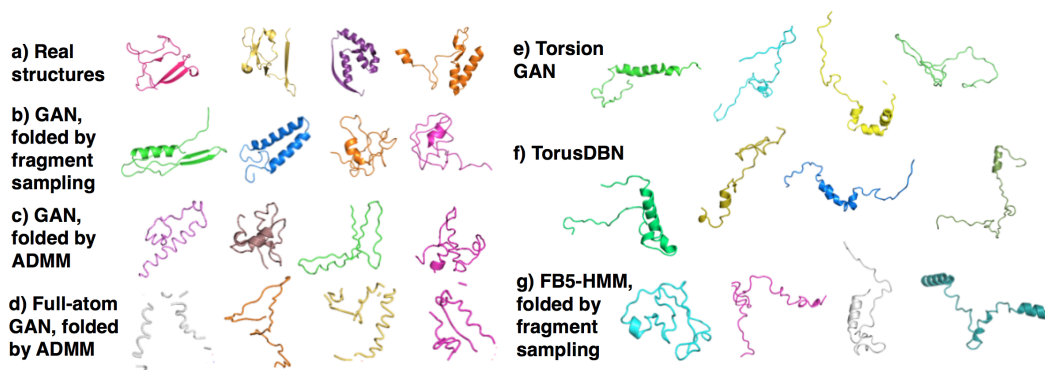

Figure 3: Examples of real 64-residue fragments **(a)** from the training dataset versus generated 64-residue fragments **(b-f)**. **b)** GAN-generated maps folded subject to $\alpha$-carbon distance constraints using Rosetta for fragment sampling. **c)** GAN-generated maps folded using ADMM and $\alpha$-carbon trace script.**d)** Full-atom GAN-generated maps folded using ADMM and $\alpha$-carbon trace script. **e)** Structures from $\phi, \psi, \omega$ angles generated by torsion GAN baseline with idealized backbone bond lengths and angles [35]. **f)** Structures generated by sampling from TorusDBN without sequence prior [17]. **g)** Structures generated by sampling $\alpha$-carbon traces from FB5-HMM without sequence prior and recovering full atom structure via Rosetta fragment sampling [18]. 3D GAN voxel output shown in Figure S8 [29].

structure (such as deviations from helix or sheet structure), while the Rosetta sampling procedure usually guarantees correct local structure.

## 4 Experiments

### 4.1 Generating protein structures

We generate 16-, 64-, and 128-residue maps and then fold them using the ADMM and Rosetta protocols above. Information on model architectures and training is given in detail in the supplementary material, Section A.1.

We compare our method for structure generation to the following baselines: Hidden Markov Model (HMM) based methods TorusDBN [17] and FB5-HMM [18], a multi-scale torsion angle GAN, 3DGAN [29], and a full-atom GAN (2D pairwise distance maps for full-atom peptide backbones). Descriptions of these baselines are given in the supplement Section A.3.

#### 4.1.1 Results

Generated maps from our trained 64- and 128-residue models are shown in Figure 2a and Figure S3a, alongside nearest neighbor maps from the training set. We found that generated maps were highly variable and similar but not identical to real data, suggesting that the GAN is in fact generating new maps and is not memorizing the training data.

We fold structures in two different ways. First, we use Rosetta to do fragment sampling subject to the generated $\alpha$-carbon distance constraints. In practice, this gives us a 3D structure with correct peptide backbone geometry that adheres to the generated constraints. Second, we use ADMM to find 3D $\alpha$-carbon placement that satisfies the generated constraints; we then use the $\alpha$-carbon trace script (described in Section 3.2.1) to trace an idealized peptide backbone geometry through the $\alpha$-carbons.

In Figure 2b, we show the distribution of mean $\ell_2$ $\alpha$-carbon map errors due to reconstruction via ADMM, the $\alpha$-carbon retrace step, and fragment sampling subject to distance constraints by Rosetta. The errors are smaller than those corresponding to nearest neighbors in the training set, and the reconstructed maps retain qualitative features of the generated maps. The $\alpha$-carbon rigid body alignment errors between the coordinate recovery methods are also small relative to nearest neighbors in the training set (Figure 2c).

In Figure 2d, we show the relative contribution to the recovered map error from the coordinate recovery process versus the generative model. While the reconstruction process introduces errors, the error is only slightly higher for the reconstruction of generated maps versus real maps for the 64- and 128- residue models. This indicates that the most of the reconstruction error is inherent to the reconstruction, versus correcting errors in the generated maps. In contrast, for a weaker 256-residue GAN, which fails to produce realistic maps or corresponding structure, the map reconstruction error for generated maps far exceeds that of real maps.

Folded structures are shown in Figure 3. We found that the generator was able to learn to construct meaningful secondary structure elements such as alpha helices and beta sheets. The Rosetta folding procedure is slow but produces locally correct structures (Figure 3c). In contrast, the ADMM folding procedure is fast but cannot correct for errors in local structure (Figure 3c). Linear interpolations in the latent space of the GAN map to smooth interpolations in the generated pairwise distances, as well as in the corresponding structures (Figures S4, S5). In addition, we found that the generator could produce maps closely corresponding to real protein structure maps through optimization of the input latent vector, indicating that the models are sufficiently complex (Section A.4, Figure S6).

Assuming average bond angles and bond lengths for the peptide backbone, the 3D structure of the backbone can be exactly represented in terms of the torsion, or dihedral, angles $\phi, \psi$, and $\omega$, defined as the angles around the $N - C_\alpha$, $C_\alpha - C$, and $C - N'$ bonds, respectively. $\omega$ is typically 180 degrees (trans orientation), except in the rare case when $\omega$ is around 0 degrees (cis orientation). The $\phi, \psi$ distribution ( Ramachandran plot) indicates allowable backbone torsion angles and different regions of the distribution correspond to alpha helix, beta sheet, and loop regions. We show the $\phi, \psi$ distribution of the generated structures and baselines in Figure S7.

The baselines (Figure 3d-g) underperform relative to the $\alpha$-carbon distance map method. Out of all the methods, the torsion GAN best adheres to the true $\phi, \psi$ distribution (Figure S7h); however, the torsion GAN, TorusDBN, and FB5-HMM baselines generate many structures which loop in on themselves and do not have realistic global 3D protein structure (Figure 3e-g).

The full-atom GAN (Figure 3d) also underperforms relative to the $\alpha$-carbon distance map method. The generated structures are represented with breaks in the peptide chain, because often the placement of backbone atoms is far enough from real backbone geometry such that the structure cannot be rendered as a contiguous chain. It is possible that a full-atom method would work better with a generative model that is multi-scale, learning both local peptide structure and global 3D structure. Results for the 3DGAN baseline are given in Figure S8; we found that this method could not generate meaningful structures.

## 4.2 Inpainting for protein design

Next, we considered how to use the trained generative models to infer contextually correct missing portions of protein structures. This is a protein design problem which arises in the context of loop modeling, circular permutation, and interface prediction. We can formulate this problem naturally as an inpainting problem, where for a subset of residues all pairwise distances are eliminated and the task is to fill in these distances reasonably, given the context of the rest of the uncorrupted structure.

We used a slightly modified version of the semantic inpainting method described in [36], omitting the Poisson blending step. This method is described in detail in the supplement Section A.5. We only present inpainting results on structures in the test set, which the GAN has not seen during training.

### 4.2.1 Baselines and metrics

We baseline with the following methods:

**10-residue supervised autoencoder.** We train an autoencoder to reconstruct completed 64-residue pairwise distance maps given input maps with random 10-residue corruptions. The encoder and decoder networks are equivalent to the discriminator and generator networks for the GAN, omitting the last layer of the discriminator and the first layer of the generator. As before, the inputs are normalized and the outputs are forced to be positive and symmetric. The autoencoder is trained with supervised $\ell_2$ loss with respect to the uncorrupted map.

**Random corruption supervised autoencoder.** We also train the same autoencoder model to reconstruct completed 64-residue maps given input maps with random corruptions ranging from 5 to 25

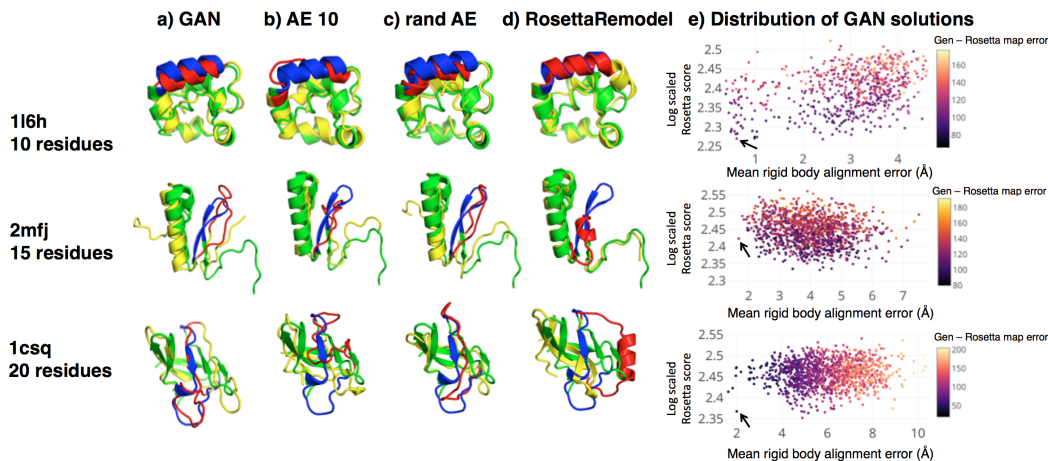

Figure 4: **GAN vs. baselines for inpainting** 10, 15, and 20 residue segments on 64-residue structures. Native structures are colored green and reconstructed structures are colored yellow. The omitted regions of each native structure are colored blue, and the inpainted solutions are colored red. **a) GAN** – solution found by sampling 1000 structures and selecting structure with low Rosetta score and low $\alpha$-carbon rigid body alignment error with respect to the native structure **(e)**. **b) Supervised autoencoder** trained on 10 residue corruptions. **c) Autoencoder trained on random corruptions** from 5 to 25 residues in length. **d) RosettaRemodel** – best Rosetta score structure after 4000 sampling trajectories. **e)** Log scaled Rosetta backbone score ($\log$(Rosetta score + 500)) vs. mean $\alpha$-carbon rigid body alignment error of the inpainted region for GAN solutions ($n = 1000$). Solutions are colored by map recovery error of Rosetta with respect to generated map. Arrows indicate rendered solution (a).

residues in length. This is to more fairly compare against the GAN inpainting method, which can handle arbitrary length corruptions.

**RosettaRemodel.** We also baseline the GAN inpainting method with RosettaRemodel [37], which uses fragment sampling to do loop closure, followed by a sequence design process, guided by a heuristic energy score. For our experiments, we do 4000 sampling trajectories per structure and rank the output solutions by their Rosetta score.

There is no canonical evaluation metric for loop closure. We can try to more quantitatively describe the correctness of the inpainting procedure using as metrics the discriminator score and the least-squares rigid-body alignment error with respect to the true native structure. However, the GAN solutions are found by explicitly trying to optimize the discriminator score; hence the corresponding score will be artificially inflated relative to the baselines. In addition, in the case where the inpainted solution is plausible but deviates from the native structure, the rigid-body alignment error will be high. Therefore, we cannot necessarily view these metrics as strong indicators of whether the reconstructed solutions are reasonable, only as rough heuristics. The ultimate test of the inpainted solutions is to experimentally verify the structures.

### 4.2.2 Inpainting results

Results for inpainting of missing residues 128-residue maps are shown in Figure S9. We see that the trained generator can fill in semantically correct pairwise distances for the removed portions of the maps. To test whether these inpainted portions correspond to legitimate reconstructions of missing parts of the protein, we fold the new inpainted maps into structures.

We render some GAN and baseline inpainting solutions for 64-residue structures folded by fragment sampling in Figure 4. We sample 1000 inpainting solutions using the generator, and render solutions with low Rosetta backbone score and low rigid body alignment error with respect to the native structure. For the Rosetta score, we only include backbone energy terms, excluding those terms involving side-chain interactions. By sampling many solutions for a single inpainting task, we can see whether native-like solutions are discoverable. In order to show that these solutions are truly

generated by the GAN and not simply found due to the fragment sampling used in map reconstruction, we color the points by the $\ell_2$ map error between the generated map and the reconstructed map after coordinate recovery.

In general, the 10-residue supervised autoencoder produces unrealistic solutions when made to inpaint longer regions; however, the autoencoder trained on random corruptions tends to do better. The primary advantages of the GAN over the autoencoder are that the GAN can handle arbitrary length loop closures without issue and that the GAN can be used to sample multiple solutions for each inpainting task, as shown in Figures S10b, S11, and S12.

While the generator and autoencoders can find inpainting solutions in minutes, including the coordinate recovery step, RosettaRemodel takes much longer. For example, for 64-residue structures, running 100 Rosetta sampling trajectories on a node with 16 CPU cores takes on average 20 minutes. For our experiments, we ran 100 trajectories per node across 40 nodes, which corresponds to about 13 hours of total CPU time per structure. It is also important to note that since Rosetta samples native fragments, it is possible for it to find the correct native solution in the course of sampling.

Discriminator score and mean rigid-body alignment error for the inpainting solutions are given in Figure 5. The mean alignment error is calculated for $\alpha$-carbons over the inpainted region only. For the GAN, results are given for 10 solutions per structure ('GAN all'), as well as for the top solutions per structure under the discriminator score ('GAN'). For Rosetta, results are given for the top 40 structures out of 4000 sampling trajectories ('Rosetta all') under the Rosetta score, as well as for the best solution among all trajectories ('Rosetta').

As the size of the inpainting region increases, the autoencoder discriminator score reduces and the structural solutions are also qualitatively worse for the autoencoder trained on 10-residue corruptions; this indicates that supervised autoencoders are not flexible enough to handle loop closure problems of arbitrary length. The GAN discriminator score is explicitly optimized during inpainting and is therefore high. In general, in terms of alignment error, the GAN inpainting solutions deviate the most from the native structures relative to the baselines.

The GAN can also be used to model longer or shorter fragments relative to the native structure; we show two examples in Figure S10c. In Figure S11 and S12, we show more GAN inpainting solutions. We are able to recover native-like solutions in terms of rigid-body error and secondary structure assignment for many inpainting tasks. There are also low-energy, non-native solutions found in a few cases. We also render some non-native and clearly implausible inpainting solutions in Figure S13.

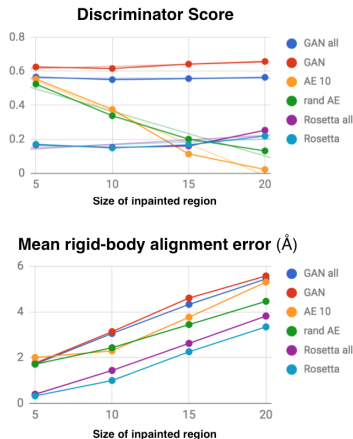

Figure 5: Discriminator score and mean coordinate $\ell_2$ alignment error for 64-residue inpainting task. Each point is averaged over $n = 64$ data-points, except for 'GAN all' ($n = 640$) and 'Rosetta all' ($n = 2560$).

## 5 Conclusion

We use GANs to generate protein $\alpha$-carbon pairwise distance maps and use ADMM to "fold" the protein structure. We apply this method to the task of inferring completions for missing residues in protein structures.

Several immediate extensions are possible from this work. We can assess whether the learned GAN features can improve performance for semi-supervised prediction tasks. We can also see how conditioning on sequence data, functional data, or higher-resolution structural data might improve structure generation [38].

Furthermore, we can extend our generative modeling procedure to solve the structure recovery problem end-to-end and mitigate the issue our current model has with making errors in fine local structure. The current approach factors through the map representation, which overconstrains the recovery problem. By incorporating the ADMM recovery procedure as a differentiable optimization layer of the generator, we can potentially extend the models presented to directly generate and evaluate 3D structures.

**Acknowledgments**

We would like to thank Frank DiMaio for helpful discussion on Rosetta and providing baseline scripts to fold structures directly from pairwise distances.

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
