[Supplementary Material]

# A Supplementary Information

## A.1 Dataset and Experiment Details

Datasets are listed in Table S1 below.

Table S1: Datasets

| Dataset | Data Size/ Type | Train data | Test data |
|---|---|---|---|
| PDB structures | input protein structures | 115850 | 6248 |
| 16 GAN | $\alpha$-carbon maps ($16 \times 16$) from 16-residue fragments | 1877174 | 85000 |
| 64 GAN | $\alpha$-carbon maps ($64 \times 64$) from 64-residue fragments | 427659 | 16329 |
| 128 GAN | $\alpha$-carbon maps ($128 \times 128$) from 128-residue fragments | 185104 | 6000 |
| 64 Full atom GAN | full peptide backbone maps ($256 \times 256$) from 64-residue fragments | 427659 | 16329 |
| 64 Torsion GAN | $\phi, \psi$, and $\omega$ angles ($3 \times 64$) from 64-residue fragments | 427659 | 16329 |

Our DCGAN model architecture is shown in Figure 1c. For all our models we used a fixed noise vector size of 100 units. Many of our models show inherent instability, but usually only after converging to a good solution for map generation. While we implemented various methods for stabilizing GAN training [39, 40, 41, 42], we found that in practice for this problem, these were not necessary for training a good model. Since we have several models in this paper, we include a list of all model architectures in the attached text file.

For upsampling by the generator, we use strided convolution transpose operations instead of pixel shuffling [43] or interpolation, as we found this to work better in practice. We set the slope of the LeakyReLU units to 0.2 and the dropout rate to 0.1 during training. Importantly, we did not normalize input maps but scaled them down by a constant factor. During training, we enforce that $G(\mathbf{z})$ be positive by clamping output values above zero and symmetric by setting $G(\mathbf{z}) \leftarrow \frac{G(\mathbf{z})+G(\mathbf{z})^T}{2}$ before passing the generated map to the discriminator. We train our models using the Adam optimizer ($\beta_1 = 0.5, \beta_2 = 0.999$) with generator and discriminator learning rates set to $10^{-4}$ and no weight decay [44]. All models were implemented using PyTorch [45].

For folding the maps using ADMM, we typically run ADMM with $\rho = 100$ and $\lambda = 1$, and run the gradient descent inner loop for 100 iterations per dual update step with learning rate $10^{-4}$; we break when the change in the primal residual is less than $10^{-2}$.

## A.2 Corruption robustness of ADMM

In Figure S2, we take maps corresponding to real structures and randomly corrupt a fraction $m$ of distances with noise sampled from $\text{Unif}(-c, c)$, while varying the slack weight $\eta$ and the Lagrangian penalty weight $\rho$. The error is calculated by doing least-squares rigid-body alignment of the new coordinates with respect to the coordinates for the true structure.

We see that for $c = 5, 10$, the rigid-body alignment error is roughly constant until about 10% of the pairwise distance measurements are corrupted. Note that the pairwise distances are of order $\sim 10\text{Å}$.

## A.3 GAN baselines

We compare our method for structure generation to the following baselines:

**TorusDBN**. The TorusDBN model is an HMM with a 55-dimensional hidden state, with emission distributions over torsion angles ($\phi, \psi$), amino acid sequence, secondary structure, cis/trans conformation ($\omega$), and NMR chemical shifts. We sampled $\phi, \psi$, and $\omega$ angles from the model without conditioning on sequence or secondary structure [17].

**FB5-HMM**. The FB5-HMM model is a generative model for $\alpha$-carbons, where the $\alpha$-carbon coordinates are defined by psuedo bond angles and dihedral angles defined for consecutive $\alpha$-carbons. We sample bond angles and dihedral angles from the model and fold full atom structures using Rosetta fragment sampling as well as our $\alpha$-carbon trace script [18].

**Torsion angle GAN.** We train a multi-scale DCGAN to generate 64-residue length $\phi, \psi$, and $\omega$ angles for protein fragments. The model architecture is given in the supplementary material. We include discriminators for 1, 2, 4, 8, 16, and 64 length scales.

**3DGAN – voxel space deep convolutional GAN.** Current methods for 3D structure generation use Cartesian or voxel-space representation of structures. We implement the 3DGAN model [29] which is a DCGAN with 3D convolutions. We transform full atom peptide backbone coordinates to voxel-space by translating the center of mass to the origin and then linearly interpolating along atomic bonds. We augment the data with random rotations about the origin.

**Full-atom GAN.** Finally, we train our 2D pairwise distance map GAN on full-atom peptide backbones (including backbone nitrogen, carbon, and oxygen atoms in addition to $\alpha$-carbons). This quadruples the size of the input maps. Importantly, for a real structure's map the ADMM coordinate recovery step has two optimal solutions– the true coordinates and the reflection of those coordinates about the origin. For $\alpha$-carbon coordinate recovery, we found that reflection of the coordinates did not lead to implausible final structures; however, for full atom solution, we check the $\phi, \psi$ distribution to determine if the coordinates are likely reflected after coordinate recovery.

## A.4 Complexity of the GAN

To test the complexity of the generative model, we asked whether for test native structure $\mathbf{x}$ we could find a corresponding $\mathbf{z} \in \mathbb{R}^n$ such that $G(\mathbf{z}) \approx \mathbf{x}$. To do this, we optimized $\mathbf{z}$ using pretrained GANs with a modified reconstruction loss objective $L_z$, adding a K-L divergence regularizer term $L_{KL}$ over the mean and variance of elements of $\mathbf{z}$.

$$L_z(\mathbf{z}) = (\|G(\mathbf{z}) - \mathbf{x}\|_2) + \gamma \, L_{KL}(\mathbf{z}) \tag{5}$$

$$L_{KL}(\mathbf{z}) = \frac{1}{2}(1 + \log(\sigma^2) - \mu^2 - \sigma^2) \tag{6}$$

where

$$\mu = \frac{1}{n}\sum_{i=1}^{n}\mathbf{z}_i, \quad \sigma^2 = \frac{1}{n}\sum_{i=1}^{n}(\mathbf{z}_i - \mu)^2$$

In practice, we set $\gamma = 10$ and optimize $\mathbf{z}$ with Adam ($\beta_1 = 0.5, \beta_2 = 0.999$) with learning rate $10^{-2}$ for 3000 steps, reducing the learning rate by 3% every 10 steps. Results for recovery of 64-residue and 128-residue structures are shown in Figure S6. We successfully recover maps with most of the input structural details; this suggests that GAN latent space encodes maps corresponding to unseen protein fragments. For the 128-residue maps, occasionally details are lost in the recovered map, which suggests perhaps we need to increase the complexity of that model.

## A.5 Inpainting objective

We used a slightly modified version of the semantic inpainting method described in [36], omitting the Poisson blending step. This method involves optimizing the input vector $\mathbf{z}$ of the GAN to find a fake image which, when overlayed over the masked region of the input, gives a good inpainting solution. There are three loss terms optimized for this procedure. The first is a context loss term, which is an $\ell_1$ reconstruction loss with higher weighting for pixels nearer to the masked region of the input. Given input $x$ and binary mask $M$ delineating the area to be inpainted, the weighting term $W$ is found by convolving the mask complement $M^C$ with a 2D identity filter of fixed size. For our experiments with 64-residue and 128-residue maps, we set the filter sizes to $9 \times 9$ and $15 \times 15$, respectively. The context loss is

$$L_{\text{context}}(\mathbf{z}) = \|(W * M^C) * (G(\mathbf{z}) - \mathbf{x})\|_1 \tag{7}$$

The next loss term is a prior discriminator loss with respect to the generated image used for the inpainting.

$$L_{\text{prior}}(\mathbf{z}) = \log(1 - D(G(\mathbf{z}))) \tag{8}$$

Finally, there is the discriminator loss on the final inpainting solution.

$$L_{\text{disc}}(\mathbf{z}) = \log(1 - D(M * G(\mathbf{z}) + M^C * \mathbf{x})) \tag{9}$$

The full objective is

$$\min_{\mathbf{z}} \quad L_{\text{context}}(\mathbf{z}) + \lambda \, L_{\text{prior}}(\mathbf{z}) + L_{\text{disc}}(\mathbf{z}) \tag{10}$$

where we set weighting term $\lambda = 0.003$. We optimize $\mathbf{z}$ with Adam ($\beta_1 = 0.5, \beta_2 = 0.999$) with learning rate $10^{-1}$ for 2000 steps, reducing the learning rate by 5% every 10 steps. We enforce that the generator output $G(\mathbf{z})$ be positive and symmetric.

# B Supplementary Figures

Figure S1: Runtimes for coordinate recovery methods (seconds) vs. protein length. Each point is an average over 64 trials. ADMM (**b**) and the $\alpha$-carbon trace step (**c**) scale linearly with problem size, while Rosetta fragment sampling scaling is approximately quadratic (**a**). Although SCS is fast for small problem sizes with linear runtime (**d**), for protein lengths larger than 115, SCS does not converge for either real or generated maps [33, 34]. ADMM and SCS convergence tolerances are $10^{-3}$ and $10^{-5}$, respectively

Figure S2: ADMM corruption robustness analysis for real protein structures. Mean rigid-body alignment error of structures increase with log fraction of corruptions $\log m$ in pairwise distances for varying slack weight $\lambda$ and Lagrangian penalty weight $\rho$. Noise added to pairwise distance matrices is $\sim\text{Unif}[-c, c]$; the dashed line represents the 10% corruption boundary. Plots show mean values for 20 examples.

Figure S3: **a)** Generated pairwise distance maps for 128-residue model, along with corresponding nearest neighbors (NNs) by $\ell_2$ distance in training dataset and maps after ADMM coordinate recovery, subsequent $\alpha$-carbon retrace step, and coordinate recovery by Rosetta fragment sampling. **b)** Distribution of $\ell_2$ map errors after coordinate recovery for generated maps ($n = 500$). **c)** Distribution of $\alpha$-carbon rigid body alignment errors between folding methods for generated maps ($n = 500$).

Figure S4: Linear interpolation in latent vector space of the generator corresponds to smooth interpolation of generated 64-residue maps.

Figure S5: Example of structure interpolation using the generator. **a)** Linear interpolation between generated maps. **b)** Corresponding structures folded by Rosetta. **c)** DSSP secondary structure assignments of interpolated structures [46, 47]. **d)** $\phi, \psi$ distribution of all interpolated structures.

**Real and recovered 64-residue maps**

**L2 map error (5.08 ± 1.7 Å)**

**Real and recovered 128-residue maps**

**L2 map error (6.64 ± 1.4 Å)**

Figure S6: Recovery of maps for 64-residue (top) and 128-residue (bottom) models by optimization of generator input vector **z** using reconstruction loss with K-L distance regularization term. Mean $\ell_2$ error for recovered maps are $5.07 \pm 1.7$ Å and $6.64 \pm 1.4$ Å for the 64- and 128-residue models, respectively ($n = 512$).

a) Real distribution    b) ADMM + CA retrace    c) Rosetta    d) Rosetta -- interpolated

$\psi$

e) TorusDBN    f) FB5-HMM + CA retrace    g) FB5-HMM + Rosetta    h) Torsion GAN

$\phi$

Figure S7: Ramachandran plot ($\phi, \psi$ distribution) for real proteins, baselines, and interpolated structures ($n = 100$). Full atom GAN generated structures excluded due to unnatural chain breaks.

Figure S8: Examples of real 64-residue fragments from the training set **(a)** versus 3DGAN generated 64-residue fragments **(b)**. Input structures are represented with full-atom peptide backbones in voxel space (1 cubic angstrom per voxel).

Figure S9: Examples of inpainting for 20 missing residues on 128-residue maps. (From left to right) Original uncorrupted input data; masked input data corresponding to deletion of all pairwise distances for 20 consecutive residues; fake sample generated by model to fill in masked region; final inpainted solution.

Figure S10: **a)** Examples of 20-residue inpainting solutions for 128-residue structures folded using ADMM (PDB ID listed under structure). Sheet-like (top) and helix-like (bottom) solutions given. **b)** Discovery of multiple solutions by randomly initializing latent vector before optimization (10-residue inpainting for 64-residue structures) **c)** GAN inpainting allows for modeling of longer or shorter segments (64-residue structures): inpainting native 10 residue segment with 15 residue solution (left); inpainting native 15 residue segment with 10 residue solution (right). Native structures are colored green and reconstructed structures are colored yellow. The omitted regions of each native structure are colored blue, and the inpainted solutions are colored red

Figure S11: **Distribution of inpainting solutions for 64-residue structures.** Inpainted maps are folded using Rosetta. **Left:** Log scaled backbone Rosetta score ($\log$(Rosetta score + 500)) vs. rigid body error with respect to native structure; points colored by L2 map error between generated map and recovered map after folding. **Right:** Inpainting solutions corresponding to blue, green, and red points indicated, along with DSSP secondary structure assignments for inpainted region (H – helix, E – sheet, L – loop) [46, 47].

Figure S12: **Distribution of inpainting solutions for more 64-residue structures.** Inpainted maps are folded using Rosetta. **Left:** Log scaled backbone Rosetta score $(\log(\text{Rosetta score} + 500))$ vs. rigid body error with respect to native structure; points colored by L2 map error between generated map and recovered map after folding. **Right:** Inpainting solutions corresponding to blue, green, and red points indicated, along with DSSP secondary structure assignments for inpainted region (H – helix, E – sheet, L – loop) [46, 47].

Figure S13: Examples of non-native and incorrect inpainting solutions for selected 128-residue and 64-residue structures, respectively, folded using ADMM (PDB ID listed under structure). Native structures are colored green and reconstructed structures are colored yellow. The omitted regions of each native structure are colored blue, and the inpainted solutions are colored red.