[Reviews · NeurIPS 2018]

Reviewer 1



The authors used GANs and convex optimization for generating and completing fixed-length protein structures. They used GANs to first generate protein residue distance maps, which they then folded into three-dimensional structures more efficiently than existing methods by convex optimization (ADMM). They compared their approach to indirectly generate protein structure via the generation of distance maps to alternative approaches such as using GANs to directly generate three-dimensional protein structures. As far as I know, this is the first paper on generating protein structures usings GANs. The paper is well written and methods clearly explained and evaluated. Provided that the authors address my comments below, I think this a good paper for NIPS. Major comments ============= 1) The authors should clarify how they sampled latent factors z that are used as input to the generator G to produce distance maps G(z). What is the variation of protein structures depending on the variation in z? Are all sampled protein structures stable (plausible)? To analyze this, the authors should perform a latent factor interpolation as mentioned in the discussion and also analyze if latent factors z capture certain properties, like the secondary structure or protein accessibility. 2) Section 4.1.1: For analyzing overfitting, the authors should also compare 2d density maps of training, test, and generated amino-acid distance maps (https://seaborn.pydata.org/generated/seaborn.kdeplot.html). 3) The authors should compare the runtime of ADMM and Rosetta depending on the protein size and add a figure with the protein size on the x-axis and observed runtime in seconds on the y-axis. 4) The method is in my eyes only of limited use for protein design since it only enables the generation of relatively short protein fragments of up to ~128 residues, is not conditioned on certain target properties like binding affinity, and does not reveal the protein or DNA sequence that folds to the generated structure. Generated structures are also of minor quality compared to structures generated by the state-of-the-art method for protein design Rosetta. The authors should therefore less emphasize the usability of their method for protein design throughout the manuscript, e.g. in the abstract. 5) The authors should more clearly discuss the drawbacks of their method in the discussion, including that their method 1) only enables the generation of fixed-length sequences, 2) is hard to scale to large protein structures, 3) requires still relatively costly and only approximately correct convex optimization since it does not directly generate three-dimensional structures, 4) learning GANs is difficult as pointed out in the suppl., and 5) that their method only generates protein backbones instead of amino-acid sequences. Minor comments ============= 6) L.13-15: The sentence should be reformulated or dropped since the authors did not should experimentally that generated structures are stable or functional. 7) The authors should more clearly motive in the introduction why protein completion is important. 8) Section 2.1: The authors should mention that proteins with certain properties can also be designed by directly generating amino acid or nucleic acid sequences instead of structures. In particular, the authors should reference http://arxiv.org/abs/1712.06148. 9) Section 4.2: The authors should more clearly describe how distance maps were masked for generating AE. Were distances (pixels) randomly dropped out (set to a special value), or did they randomly drop out fixed-length fragments? The latter approach is required for a fair comparison. Did the authors add noise to the dropped out regions or set them to a special value? 10) Throughout the manuscript: What does SE(3) mean? 11) L122-124: The authors should more clearly describe if they trained three GANs for generating sequences of size 16, 64, and 128. The fact that their model does not enable training variable-length sequences is one major drawback (see comment above). 12) In addition to Figure 3., the authors should show more structures generated by latent factor interpolation (see comment above) in the suppl..

Reviewer 2



The paper introduced a generative model for protein structure that is not conditioned on the amino acid sequence. The model is based on deep convolutional generative adversarial networks (DCGANs) and the output of the GAN is a pairwise intramolecular distance matrix for the C_alpha trance. The authors point out that this output representation is SE(3) invariant. Secondly, the authors develop a convex approach for reconstructing the C_alpha positions from the distance matrix based on the Alternating Direction Method of Multipliers (ADMM) algorithm. They propose using Rosetta based fragment assembly with 5-mer to convert the C_alpha trace into a full backbone representation. Finally, the authors apply the proposed model to predict missing parts of protein structures (denoted "inpainting for protein design"). Most of the evaluations are qualitative, hower for the inpainting the performance is evaluated in terms of discriminator score and mean rigid-body alignment error. Measured by the discriminator score the proposed DCGAN model outperforms the other models, but measure by mean rigid-body alignment error the DCGAN is outperformed by the other models/methods (with a margin of about 2Å). This work represents a very interesting application of GANs that goes beyond the usual domain of images. The manuscript is well written and easy to follow, and the work appears scientifically sound. However, this DCGAN model itself (for the distance matrix) is a straightforward application of DCGANs without any significant methodological contributions. The idea of using an SE(3) invariant matrix represent is also not new. For sequence dependent prediction, there is a large body of literature on predicting the contract matrix, see e.g. Marks et al. [Nature Biotechnology, 2012], and very recent work on supervised learning of protein structure with the distance matrix as the target [AlQuraishi, bioRxiv 265231, 2018]. Even though the papers mentioned above are sequence dependent work and the proposed method is sequence-agnostic, the relation to this previous work should be discussed/mentioned in the "related work" section. The authors mention that they are not aware of any prior work for generating protein structure without condition on sequence information. Some prior work has been done in this direction: the models FB5-HMM (Hamelryck et al., PLOS Computational Biology, 2006) and TorusDBN (Boomsma et al., PNAS, 2008) are generative models of protein sequence and local structure. Since they are generative (modelling the joint distribution of sequence and structure) they can of cause be used without condition on sequence information. The proposed solution to the reconstruction problem (reconstruction of the C_alpha trace from the distance matrix) using ADDM is an interesting contribution from a protein modelling perspective. However, the contribution is very incremental and it is not part of the learning algorithm. Also, the construction of the full backbone from the C_alpha trace using Rosetta fragment sampling of 5-mers seems very ad-hoc and is not part of the learning algorithm. The evaluation of the proposed method is very qualitative (which the authors themselves also points out). By visual inspection of figure 3, the authors conclude that the qualitative evaluation shows that the DCGAN+Rosetta generates fragments that are folded-like and have realistic secondary structure, while the DCGAN+ADDM generates folded-like fragments that do not have realistic secondary structure. The proposed baseline models do not generate realistic structures. Here, it would be relevant to do a comparison with an existing model from the literature, such as TorusDBN. For the prediction of missing parts of protein structures, the authors perform a quantitative evaluation by measuring the discriminator score and mean rigid-body alignment error. Here, I see the main conclusion to be that the DCGAN performs about 2Å worse than Rosetta in mean rigid-body alignment error, but DCGAN only has a significant lower computation cost than Rosetta. I would be highly relevant to also look at the error measured in DRMSD in order to access how much of the error is due to the generative model and how much of the error is due to the reconstruction. The paper is clearly written and the scientific work is of high quality. However, in its present form, the contribution/significance of the paper is somewhat limited as the generative model is a straightforward application of DCGANs. If the ADMM based reconstruction procedure was incorporated as part of the model (as proposed by the authors as future work), I would see view this as more significant and original contribution. Rebuttal: The authors have addressed some of my comments, and in particular, they intend to add the baselines mentioned above. I am happy to see that the authors will report the lowest error of the GAN solution. However, as no additional result are not included in the rebuttal, this additional work cannot be evaluated. In conclution, I maintain my recommendation.

Reviewer 3



Overview In silico protein folding is an essential part of many bioengineering pipelines; however, all existing methods are highly computationally expensive, which makes the generation of large numbers of candidate protein structures infeasible. The authors propose an alternative to the incumbent biophysical models (the most common of which is Rosetta) which uses instead a generative adversarial network (GAN) to generate a pairwise distance matrix for alpha-carbons within a variable-length amino acid chain. The authors then use an existing algorithm called Alternating Direction Method of Multipliers (ADMM) to reconstruct the 3D position of the alpha-carbons, and finally use a partial solution from Rosetta to obtain the full amino acid backbone structure. Technical quality The authors’ method is clearly and adequately described. The experiments are well thought out and results are presented fairly. In the de novo protein folding experiment, the authors compare their method to five other protein folding methods. Aside from 3DGAN (Wu et al, NIPS 2016), it is not clear which (or any) of these are published methods, or whether they were partial methods invented by the authors. Regardless of the origin of these methods, the authors show that potential alternative formulations of their method using torsion angles, full-atom GANs, or Hidden Markov Models are insufficient for de novo protein generation. However, between the GAN folded by fragment sampling and the GAN folded by ADMM, it is worth noting that none of the proteins generated by the authors’ chosen method (ADMM) contain any beta sheets. This pattern is clearly repeated in the inpainting experiment, where not one of the proteins in which the original inpainted section contained a beta sheet (1mva, 3h3i, 4ang, 1ahk, 3ame) has a beta sheet when inpainted by ADMM GAN. As one of the most common tertiary amino acid structures, the failure to recapitulate a beta sheet is a severe technical flaw in the proposed method. Additionally, as noted by the authors, the metrics used for numerical comparison of the tested methods are significantly flawed. The authors note that “in the case where the inpainted solution is plausible but deviates from the native structure, the rigid-body alignment error will be high”. However, the authors could at the very least verify that the GAN is capable of producing *a* plausible solution by examining the distribution of rigid-body alignment errors for all possible GAN inpainting solutions. If the minimum (or e.g. 10th percentile) rigid-body alignment error across hundreds of inpainting solutions approaches the scores achieved by Rosetta, this would be a convincing argument that the GAN is capable of generating biologically plausible solutions, and that other solutions with higher rigid-body alignment error are simply alternative solutions less similar to the original protein that may still be valid folded proteins. Novelty While the authors’ proposed algorithm is indeed a novel pipeline, and to date the first attempt to apply GANs to protein modeling, it should be noted that none of the component parts of the algorithm are novel. ADMM was invented in 2011 and has been previously applied to protein folding, and the semantic inpainting method used in section 4 was previously described in 2016. Thus the authors present a novel application of existing technologies. Potential impact The authors’ work has the potential to be highly impactful, contingent on the resolution of the technical failure to produce beta sheet structures. De novo protein design is a highly computationally expensive task, and additionally requires human intervention. A fast, automated and performant method to design proteins in a high-throughput manner could revolutionize drug design; however, the authors have not yet demonstrated that their algorithm is genuinely competitive with slower alternatives. Clarity The paper is well written and easy to read. Minor comments: - “Our model for the scope of this paper is not necessarily learning protein structures which will fold, but rather is learning building block features that define secondary and tertiary structural elements.” – this sentence is confusing. - The authors claim that the semidefinite program to infer 3D coordinates from pairwise distances is quadratic, and that ADMM by contrast converges “quickly”. What is quickly? Could this be shown, perhaps in a supplementary figure? - It would also be good to see the difference in performance of the ADMM folding method compared to the GAN + fragment sampling shown in Figure 3. - Equation 3 fails to define \lambda and Z. - Equation 4: should G_{k+1} be dependent on G_k, rather than G? - If is not clear how the alpha-carbon trace algorithm works, and whether it is a custom script written by the authors, or included in the Rosetta package. The authors present a novel application of GANs in a field where the impact of such a technology could be very high. (post response) The authors have addressed my major concern with the manuscript, the inability to produce beta sheets, as this inability is caused not by the ADMM GAN but by the protein rendering software which creates the cartoons. Discussion of how to overcome this issue, in addition to added reporting on runtimes and the distribution of rigid-body alignment error will make more convincing the capability of GANs to revolutionize the field of in silico protein modeling. Recommendation: weak accept.